# Engineered Microvessel for Cell Culture in Simulated Microgravity

**DOI:** 10.3390/ijms22126331

**Published:** 2021-06-13

**Authors:** Mei ElGindi, Ibrahim Hamed Ibrahim, Jiranuwat Sapudom, Anna Garcia-Sabate, Jeremy C.M. Teo

**Affiliations:** 1Laboratory for Immuno Bioengineering Research and Applications, Division of Engineering, New York University Abu Dhabi, Abu Dhabi P.O. Box 129188, United Arab Emirates; me95@nyu.edu (M.E.); ihi2005@nyu.edu (I.H.I.); jiranuwat.sapudom@nyu.edu (J.S.); anna.sabate@nyu.edu (A.G.-S.); 2Department of Mechanical and Biomedical Engineering, New York University, 6 MetroTech Center, Brooklyn, NY 11201, USA

**Keywords:** cell culture microvessel, space biology, random positioning machine, simulated microgravity

## Abstract

As the number of manned space flights increase, studies on the effects of microgravity on the human body are becoming more important. Due to the high expense and complexity of sending samples into space, simulated microgravity platforms have become a popular way to study these effects on earth. In addition, simulated microgravity has recently drawn the attention of regenerative medicine by increasing cell differentiation capability. These platforms come with many advantages as well as limitations. A main limitation for usage of these platforms is the lack of high-throughput capability due to the use of large cell culture vessels. Therefore, there is a requirement for microvessels for microgravity platforms that limit waste and increase throughput. In this work, a microvessel for commercial cell culture plates was designed. Four 3D printable (polycarbonate (PC), polylactic acid (PLA) and resin) and castable (polydimethylsiloxane (PDMS)) materials were assessed for biocompatibility with adherent and suspension cell types. PDMS was found to be the most suitable material for microvessel fabrication, long-term cell viability and proliferation. It also allows for efficient gas exchange, has no effect on cell culture media pH and does not induce hypoxic conditions. Overall, the designed microvessel can be used on simulated microgravity platforms as a method for long-term high-throughput biomedical studies.

## 1. Introduction

In recent years the number of manned space flights has increased significantly. Research has shown the importance of gravitational forces on the development and function of all living organisms and how microgravity conditions result in physiologically detrimental effects [1,2,3,4]. Thus, the study of the effects of microgravity is critical in helping us understand how organisms will be affected as space exploration continues. However, sending samples into space is a complicated and costly procedure [5]. To overcome this, less expensive platforms exist, such as parabolic flights or drop towers, but these provide very short durations of microgravity for experimentation, which is usually not adequate to replicate the long-term effects of microgravity on biological samples. Due to this, simulated microgravity studies have become more widely used in recent years [6]. Commercial platforms such as two- (2D), and three-dimensional (3D) clinostats, the rotating wall vessel (RWV), and the random positioning machine (RPM) are currently available and used to simulate the effects of microgravity on Earth [7]. These platforms have also become recently used to increase cell differentiation capability for regenerative medicine [8,9,10,11]. 

For the multitude of platforms available for simulated microgravity studies, there are associated limitations to biological studies performed using these devices. One of the primary drawbacks is the lack of small vessels to be used for biological experiments on these simulated microgravity platforms. Current procedures for biological studies primarily use relatively large vessels to carry out experiments (e.g., T-25 and T-75 cell culture flasks) [12,13,14,15,16] (Figure 1) or custom-made cell culture vessels [17,18,19]. Comparatively, these larger vessels require a copious amount of cells and cell culture media which is cost intensive, uses large amounts of single use plasticware, and limits biological studies that necessitate smaller amounts of media, for example the analysis of cytokine secretion profiles. Larger vessels also limit the number of experimental replicates (low throughput) to be run concurrently which ultimately requires more time for experimental completion with sufficient statistical relevance and will eliminate batch-to-batch discrepancies between samples. The availability of microvessels for simulated microgravity research is a vital consideration, especially if the experiment is limited by cell number (e.g., primary stem cells) and costly novel drug treatments [20].

This work aimed to design a versatile custom-made biocompatible microvessel that allows for cell-based research to be performed on the RPM, as well as other simulated microgravity platforms. It can also be further used for experimental runs on other dynamic platforms. Designing a microvessel that is compatible with smaller sample volumes would allow for more high-throughput experiments to take place, with the additional benefit of utilizing fewer biological materials.

## 2. Results and Discussion

As mentioned above, biological experiments on simulated microgravity platforms often utilize large vessels for samples which hinders the ability to perform high-throughput experiments (Figure 1). This also limits samples from being positioned equidistant from the center of the platform where centrifugal forces are negligible [21]. Due to centrifugal forces, fluid motion has been shown to appear in large vessels (e.g., T25 flasks) [22]. Hence, we would expect a lower degree of fluid motion to appear in smaller vessels [20,23]. In this work, the microvessel is designed to allow for high-throughput biological assays to be performed while limiting the use of consumables. RPM experiments using conventional large vessels and the designed microvessels are depicted in Figure 1. To successfully fabricate this easy-to-use microvessel, material biocompatibility was first assessed by cell viability and immune response assays towards the materials. Subsequently, the microvessel was designed using the suitable materials to have the optimum orifice diameter which allows for gas exchange and prevents loss of liquid/cell culture media. This microvessel was then tested for its effect on long-term cell viability, proliferation and risk for developing hypoxic conditions.

### 2.1. Selecting the Suitable Biocompatible Materials for Microvessel Fabrication

In order to determine the most suitable and biocompatible material for fabrication of the microvessel, four 3D printable (polycarbonate (PC), polylactic acid (PLA) and resin) and castable (polydimethylsiloxane (PDMS)) materials were fabricated into 13 mm disks (Figure 2A). Adherent cells (human dermal fibroblasts and MDA-MB-231 cells) and suspension cells (THP-1 and Jurkat) were cultured on top of the materials for 24 h. The material biocompatibility was assessed using DRAQ7, a live/dead assay. As shown in Figure 2B–E, all cells showed high cell viability when cultured on PC, PLA and PDMS, while cells cultured on resin exhibited significant reduction in percentage of live cells in all four cell types (Figure 2B–E). Therefore, resin was eliminated as a potential material for fabrication of the microvessel.

Besides cell viability, immunological response is a key factor to be considered in material selection. The materials used for microvessel fabrication should not cause unintended inflammatory responses. To test immunological response, pro-inflammatory cytokine secretion from THP-1 cells was analyzed after 24 h culture with the different materials. THP-1 cells are a human monocytic cell line and are an established cell model for testing a material’s immunogenicity [24]. IL-1β, TNFα and IL-18 are pro-inflammatory cytokines secreted by monocytic cells in response to foreign materials [25]. As shown in Figure 3A,C, no significant change in IL-1β and TNFα was observed in all tested materials. However, there was a significant increase in IL-18 levels when THP-1 was incubated with PC (Figure 3B). Due to the immunological response to PC, this material was not chosen for fabrication of the microvessel.

### 2.2. Design of an Easy-to-Use Microvessel

Given the viability and cytokine data obtained, PDMS and PLA were chosen as potential materials for fabrication of the microvessel. The general design of the microvessel is depicted in Figure 4A. The microvessel is designed to contain 500 μL of liquid or cell culture media. To ensure optimal gas exchange and prevent any liquid or cell culture media leakage, the critical diameter of the orifice was calculated at different gravity levels (Figure 4B). The critical orifice diameter was found to be 4.53 mm at 1 g, when the orifice is facing downward along the direction of gravity. The microvessel was designed to fit a standard 4-well plate. This critical orifice diameter remains the same if the design is adapted to fit in other standard well plates such as 24-well and 6-well plates. PDMS microvessels were cast in a custom made mold (Appendix A) with dimensions for a 4-well plate (Appendix A). PLA was printed using the same dimensions. The designed microvessel is simple, user friendly and easy to fill with liquid or media (Figure 4C). To determine the ability of the microvessel to maintain the volume of media in the wells without leakage or formation of air bubbles, liquid was placed into wells of a 4-well plate with both PLA- and PDMS-generated microvessels and was weighed before being rotated on the RPM at 60 deg/s. The plates were placed as close to the center of rotation as possible to minimize centrifugal forces. Liquids were subsequently weighed after day 1, 3 and 6 (Figure 4D). The liquid in microvessels fabricated from PLA had a 70% decrease in weight due to leakage within the first 24 h. This was most likely because of the rigidity of the material that prevents it from deforming to fit tightly into the wells. This could also be due to the non-uniform well sizes in the manufacturing process of the plates. On the other hand, PDMS provides a flexible structure that overcomes the limitation of PLA fabricated microvessels. Figure 4D shows the weight of liquid in the plates with PDMS microvessels. There was minimal decrease in weight (less than 1%) over the 6 days under rotation of the RPM, indicating that there was no loss of liquid from the microvessels through droplets, seeping or evaporation. This results in no formation of significant air bubbles which would prevent the accurate simulation of microgravity. Furthermore, air bubbles could induce forces on the cells that would influence the biological outcome; for example, they could potentially cause adhered cells to become detached from the surface [26]. In sum, PDMS is more elastic than PLA and can tightly fit to the size of the wells, despite small changes in well dimensions, creating an adequate seal to prevent loss of liquid or cell culture media. The microvessel can easily be redesigned to fit any well or vessel size.

Since biological experiments require maintenance of CO_2_ levels in the media, the orifice must allow for efficient gas exchange to occur. To determine the gas exchange ability of the PDMS microvessels, RPMI-1640 cell culture media was incubated with and without the microvessels for 6 days. No change in cell culture media color was observed (Figure 5A). To confirm the observation, change in cell culture media pH was assessed by means of phenol red using a spectrophotometer. Absorbance of phenol red was measured at a range of wavelengths (450–650 nm) and the peaks at 505 nm and 560 nm indicate the wavelength readings for acidic and basic conditions, respectively (Figure 5B). To simulate acidic and basic conditions, 2 M HCl and 0.75 M NaOH were added to cell culture media, respectively. The data show no significant difference between the absorbance of phenol red in the media contained in both well plates, with and without microvessels, for up to 6 days, indicating that there is adequate gas exchange occurring in the presence of microvessels (Figure 5B,C). These data suggest that the critical orifice diameter calculated is adequate for both gas exchange and maintenance of liquid volume in the microvessels.

To further ensure that the PDMS microvessels allow for sufficient gas exchange in cell culture conditions, *HIF1α* gene expression was measured as a marker for induction of hypoxia in cells [27]. Levels of *HIF1α* gene expression remained stable across all four cell types cultured with microvessels relative to samples cultured without microvessels (Figure 5D). These data confirmed the adequate gas exchange with no indication of hypoxic conditions in the microvessel when compared to standard cell culture conditions.

### 2.3. Microvessel Does Not Affect Cell Viability and Proliferation in Long-Term Cell Culture

As we demonstrated that the PDMS microvessel did not generate hypoxic conditions, the microvessel should allow long-term culture of cells. In order to verify long-term effects of the PDMS microvessel on cell behavior, THP-1, Jurkat, MDA-MB-231 and fibroblast cells were cultured in standard laboratory conditions, not in simulated microgravity, with and without microvessels for 6 days. Figure 6A shows no significant change in morphological appearance of all cell types after 6 days in culture. Besides this, cells were quantified regarding their viability using DRAQ7 assay. At day 1, 3 and 6 there was no significant change in cell viability of all cell types cultured within microvessels when compared to control samples without microvessels (Figure 6B–E).

Alongside the morphological and viability aspects, proliferation of cells is an important parameter to take into consideration. Similar to the viability assay, cells were cultured with and without PDMS microvessels in standard laboratory conditions. Cell proliferation was determined by counting cell numbers using flow cytometry at day 1, 3 and 6. Figure 7 shows no significant changes in cell numbers relative to day 1 samples, with and without microvessels for all 6 days. In sum, the PDMS microvessel is suitable for long-term cell culture without affecting cell viability and proliferation in both adherent and suspension cells.

## 3. General Discussion and Conclusions

Overall, the PDMS microvessel was found to be suitable for long-term cell culture based on tests for its effect on long-term cell viability and proliferation, induction of hypoxic environment, effect on media pH and ability to be used on the RPM without losing volume or creating air bubbles. In addition, due to the transparency of the PDMS, it allows in situ live cell imaging. Together, these data provide evidence that this custom-made microvessel is a unique approach to performing high-throughput biological studies with use of fewer consumables on simulated microgravity platforms. Furthermore, because part of the technology uses commercially available cell culture consumables, it facilitates seamless transition of laboratory protocols to simulated microgravity. To further confirm the presence of shear forces and fluid motion in our system, computational fluid dynamics (CFD) simulations should be performed as these may affect cell behavior. According to the manufacturers, samples within a 4 mm distance from the center of rotation have negligible centrifugal forces at 60 deg/s. The microvessels engineered in this work can be placed such that one well is at the center of rotation, and is thus as close to the 4 mm distance restriction as possible. This allows users to test the effect of different operational modes on the RPM, such as clinostat mode, one directional random speed, etc., to see which is most effective for the study at hand [23]. Furthermore, as they are fabricated out of PDMS, the microvessels can be modified to add ports for microfluidics which will greatly increase the application properties. Providing options for media exchange or addition of reagents, using microfluidics, without the need to stop and remove samples from the simulated microgravity platform during the dynamic experiments, will allow for more accurate results, since adaptation times to ground and microgravity levels are still being studied [28,29].

## 4. Materials and Methods

### 4.1. Cell Culture

For suspension cells: Human monocytic cell line THP-1 and human T lymphocyte cell line Jurkat were maintained in RPMI-1640 cell culture media supplemented with 10% fetal bovine serum (FBS), 1% HEPES, 1% sodium pyruvate, 0.01% beta-mercaptoethanol and 1% penicillin/streptomycin at 37 °C, 95% humidity and 5% CO_2_ (standard cell culture conditions). Cell culture media and supplements were purchased from Gibco, Invitrogen, Thermo Fisher Scientific Inc., Dreieich, Germany.

For adherent cells: MDA-MB-231 cell line and primary human dermal fibroblasts were maintained in DMEM cell culture media supplemented with 10% fetal bovine serum (FBS) and 1% penicillin/streptomycin at standard cell culture conditions. Cell culture media and supplements were purchased from Gibco, Invitrogen, Thermo Fisher Scientific Inc., Dreieich, Germany. For all experiments, cells were seeded at 2 × 10^5^ cells/well. 

### 4.2. Fabrication of Materials for Biocompatibility Test

Materials for biocompatibility tests were fabricated as a 13 mm disk. From polycarbonate (PC; Filatech 3D Printing Industries FZC, Ras Al-Khaimah, UAE) and polylactic acid (PLA; Prusa Research a.s., Prague, Czech Republic), both filaments were printed using a Prusa i3 MK3 printer (Prusa Research a.s., Prague, Czech Republic). Resin (Dental LT Clear Resin; Formlabs, Somerville, MA, USA) was fabricated using a Form 2 3D printer (Formlabs, Somerville, MA, USA). Polydimethylsiloxane (PDMS, Dow Inc., Midland, MI, USA) disks were fabricated by mixing the elastomer base and curing agent at 10:1 ratio (SYLGARD 184, Dow Inc., Midland, MI, USA). Materials were rigorously cleaned by washing with 70% ethanol and distilled water, followed by UV sterilization prior to use.

### 4.3. Test of Materials Biocompatibility and Immunogenicity

For cell viability tests, 1 × 10^5^ cells of THP-1, Jurkat, MDA-MB-231 and fibroblasts were cultured on top of the material disks for 24 h at standard cell culture conditions. For MDA-MB-231 and fibroblasts, cells were first detached with TrypLE (Gibco, Invitrogen, Thermo Fisher Scientific Inc., Dreieich, Germany). Cells were stained with DRAQ7 (dilution 1:1000 in cell culture media; Biolegend, USA) for 10 min at standard cell culture conditions. Cells were analyzed using flow cytometry (Attune NxT, Thermo Fisher Scientific Inc., Dreieich, Germany).

To study the immunogenicity of the materials, cell culture media of THP-1 were collected and the secretion of pro-inflammatory cytokines, namely IL-1β, IL-18 and TNFα, were analyzed using bead-based ELISA (Biolegend, San Diego, CA, USA). Experiments were performed in at least 4 independent replicates.

### 4.4. Design and Fabrication of Cell Culture Microvessel

Microvessels were designed using Fusion 360 (Autodesk, San Rafael, CA, USA) and DesignSpark Mechanical (RS Components and Ansys Inc., Dubai, UAE). The microvessel was initially designed to tightly fit in a well size corresponding to a 4-well plate (Cat. No 176740, Thermo Fisher Scientific Inc., Dreieich, Germany) and contain a volume of media of approximately 500 μL. The design includes an open orifice at the top to facilitate pipetting and gas exchange inside the incubator. The orifice size has been calculated to ensure that surface tension will hold the media in place, preventing any leakage (Figure 4B). The calculation was done by balancing the Laplace pressure (Δ*p_L_*) for a hemispherical cap.
(1)$$\Delta p_{L}=\sigma \left(\frac{1}{R_{1}}+\frac{1}{R_{2}}\right)=\frac{4\sigma }{D}$$
where *σ* is the surface tension, radius of the droplet *R*_1_ = *R*_2_ and *D* is the diameter of the droplet and the hydrostatic pressure (Δ*p_h_*).
(2)$$\Delta p_{h}=\rho gh$$
where ρ is the fluid density, *g* is the gravity and *h* is the height of the fluid inside the vessel. The calculated critical orifice diameter at 1g_0_ (where g_0_ = 9.81 m/s^2^) was 4.53 mm, therefore we chose a slightly smaller diameter orifice in the final design (4 mm), to compensate for any residual accelerations during RPM operation.

To fabricate the microvessel, polylactic acid (PLA) was 3D printed using a Prusa i3 MK3 (Prusa Research a.s., Prague, Czech Republic) and polydimethylsiloxane (PDMS) was casted on custom-made CNC manufactured molds (Haas VF-2TR CNC Milling Machine (Haas Automation Inc., Oxnard, CA, USA)) (Appendix A).

Details and files to reproduce the fabrication of the microvessels used in this work are available upon request.

### 4.5. Test of Liquid Volume Loss in Simulated Microgravity Platform 

The desktop RPM 2.0 (Airbus Defence and Space Netherlands B.V., Leiden, Netherlands) was placed inside a standard incubator with 37 °C and 5% CO_2_. The RPM was operated in the random speed mode with random interval and random direction at 60 deg/s to test volume loss from the microvessel. The 4-well plates with microvessels were placed in the center of the platform as seen in Figure 1. Experiments were performed in at least 4 independent replicates.

### 4.6. Assessment of Cell Viability 

Cell viability was quantified using DRAQ7 staining (Biolegend, San Diego, CA, USA), a far-red fluorescence DNA dye that only stains the nuclei of dead cells. Briefly, cells were incubated with DRAQ7 dye (dilution 1:1000; Biolegend, San Diego, CA, USA) for 10 min at standard cell culture conditions. Afterwards, cells were analyzed using a flow cytometer (Attune NxT; Thermo Fisher Scientific Inc., Dreieich, Germany). Cells negative for DRAQ7 were counted as live cells. Experiments were performed in at least 4 independent replicates.

### 4.7. Analysis of Cell Culture Media pH Using Absorption Spectrum of Phenol Red

Changes in cell culture media pH were measured by means of phenol red, which turns yellow at acidic pH and purple at basic pH. To prevent a rapid change of pH due to exposure to CO_2_ in the air, the absorbance spectrum of phenol red was read using a Synergy H1 plate reader (BioTek, Winooski, VT, USA) at wavelengths between 450 nm and 650 nm. A total of 100 μL of 2 M HCl (Sigma-Aldrich, Schnelldorf, Germany) and 0.75 M NaOH (Sigma-Aldrich, Schnelldorf, Germany) were added to the cell culture media, to simulate the acidic and basic conditions, respectively. For the quantitative analysis, the ratio of peaks at 505 nm and 560 nm were calculated. Experiments were performed in at least 4 independent replicates.

### 4.8. RNA Isolation and Gene Expression Analysis

Gene expression analysis was performed using an established protocol, as published [30]. Briefly, total RNA was extracted using TRIzol (Invitrogen, Thermo Fisher Scientific Inc., Dreieich, Germany) and chloroform (Sigma-Aldrich, Schnelldorf, Germany), and was subsequently converted into complementary DNA (cDNA) using a high-capacity cDNA reverse transcription kit (Applied Biosystems, Thermo Fisher Scientific Inc., Dreieich, Germany). The cDNA concentration and purity (the ratio of absorbance at 260 nm and 280 nm) were quantified using nanodrop (Thermo Fisher Scientific Inc., Dreieich, Germany) prior to performing gene expression analysis. The primers used in this study were synthesized from Bioneer Inc. (Daejeon, South Korea) qPCR was performed using the SYBR Green PCR Master Mix (Applied Biosystems, Thermo Fisher Scientific Inc., Dreieich, Germany). The qPCR procedure was set as follows: denaturation for 5 min at 95 °C; 45 cycles of denaturation (95 °C, 15 s), annealing under primer-specific conditions (30 s), and target gene-specific extension (30 s at 72 °C). Fluorescence signals were measured for 20 s at 72 °C. To confirm the specificity of the PCR products, a melting curve analysis was performed at the end of each run. The Beta-actin gene was used as a reference gene. Experiments were performed in at least 4 independent replicates.

The primer sequences used are found in Table 1 below.

### 4.9. Data and Statistical Analysis

Statistical significance was determined by two-way ANOVA followed by Tukey’s post hoc test using Prism 9 (GraphPad Software Inc., San Diego, CA, USA) and the level of significance was set to *p* < 0.05. Unless otherwise stated, all experiments were performed in at least three repeats and data are represented as mean ± standard deviation (SD).

## Figures and Tables

**Figure 1 ijms-22-06331-f001:**
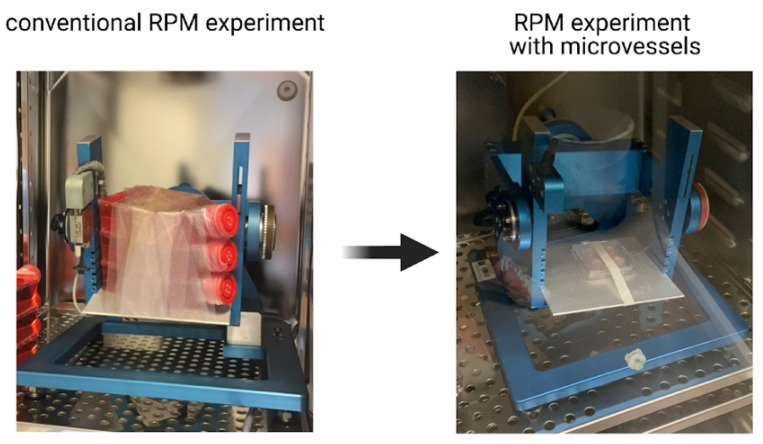
Comparison of experimental setup using conventional large vessels (**left**) and the designed microvessels (**right**). Figure depicting a conventional RPM experimental adapted from Buken et al. [13].

**Figure 2 ijms-22-06331-f002:**
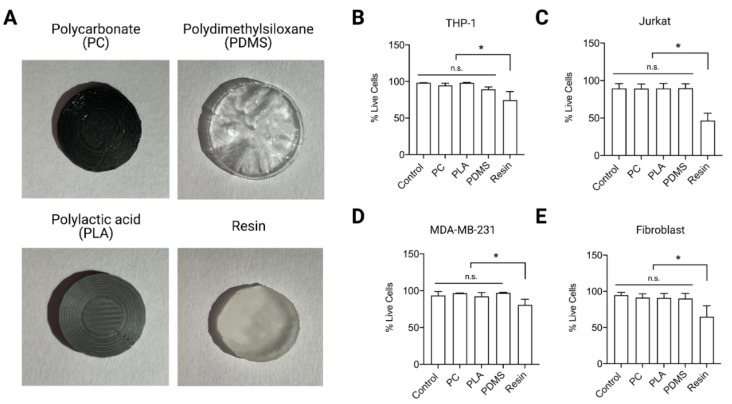
Material biocompatibility. (**A**) Representative images of selected materials for fabrication of cell culture microvessel. Cells were cultured on top of the materials (round disk; 13 mm in diameter) for 24 h. Biocompatibility of materials was elucidated using DRAQ7 staining, a commercial live/dead assay, for suspension cells (**B**) THP-1 and (**C**) Jurkat cells, and adherent cells (**D**) MDA-MB-231 and (**E**) primary human fibroblasts. Data are presented as mean ± SD. Experiments were performed in at least 4 replicates. n.s. indicates non significant. * indicates a *p* value ≤ 0.05.

**Figure 3 ijms-22-06331-f003:**
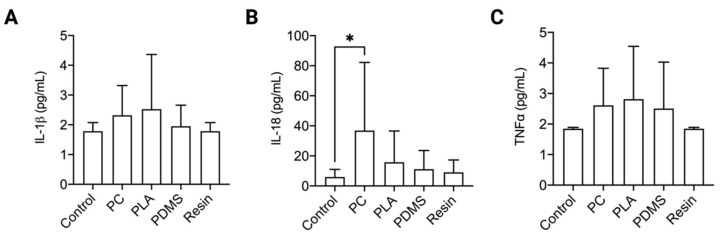
Pro-inflammatory cytokine levels. Figure shows the levels of (**A**) IL-1β, (**B**) IL-18 and (**C**) TNFα released from THP-1 cells incubated for 24 h with PC, PLA, PDMS and resin. Data are presented as mean ± SD. Experiments were performed in at least 8 replicates. * indicates a *p* value ≤ 0.05.

**Figure 4 ijms-22-06331-f004:**
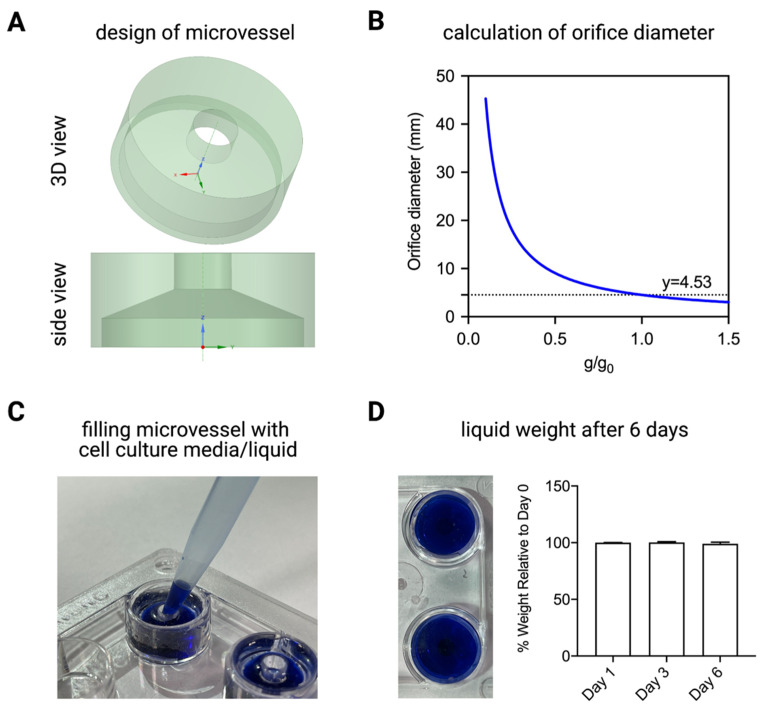
Microvessel design. (**A**) Design of microvessel seen from the top and side view. (**B**) Calculation of the orifice diameter to ensure surface tension force prevents loss of liquid or culture media from 4-well plate at different gravity levels, since the microvessels can be used on reduced gravity platforms that may also momentarily experience hypergravity. (**C**) Visual representation of the ease of filling the microvessel with culture media or liquid. (**D**) Visual representation of full microvessels with no air bubbles and graph depicting the percent of weight in the vessel after 6 days relative to day 0. Data are presented as mean ± SD. Experiments were performed in at least 3 replicates.

**Figure 5 ijms-22-06331-f005:**
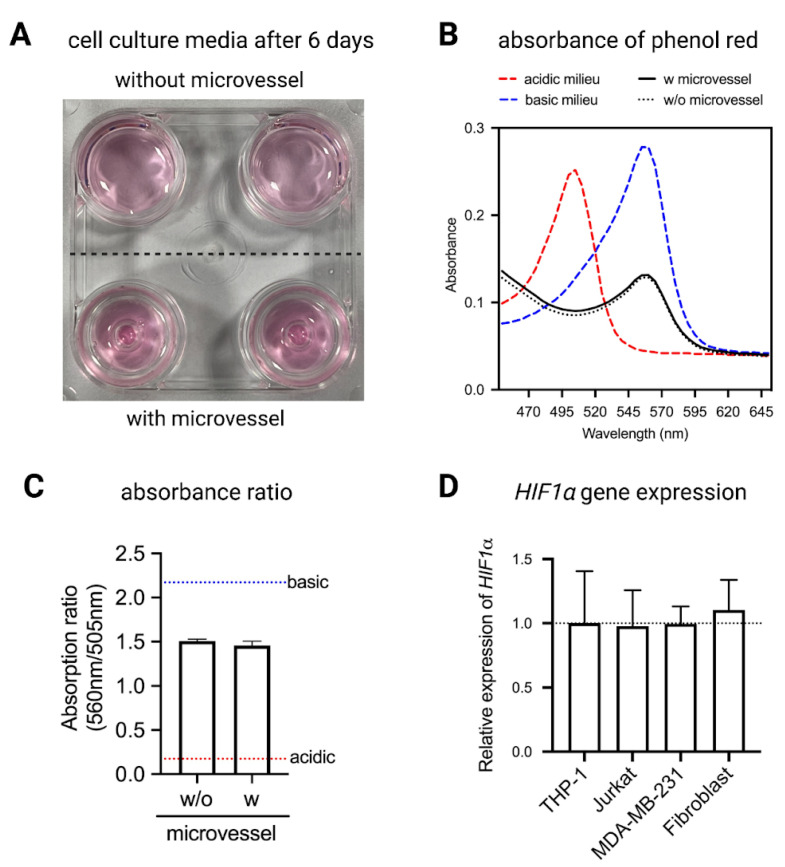
Gas exchange adequacy of microvessels. (**A**) Representative image of cell culture media after 6 days of culture with and without microvessels. (**B**) Absorbance of phenol red in cell culture media and (**C**) absorbance ratio (560 nm/505 nm) of phenol red in cell culture media. (**D**) Expression of HIF1α levels in THP-1, Jurkat, MDA-MB-231 and Fibroblast cells cultured with PDMS microvessels relative to control samples cultured without microvessels. Data are presented as mean ± SD. Experiments were performed in at least 4 replicates.

**Figure 6 ijms-22-06331-f006:**
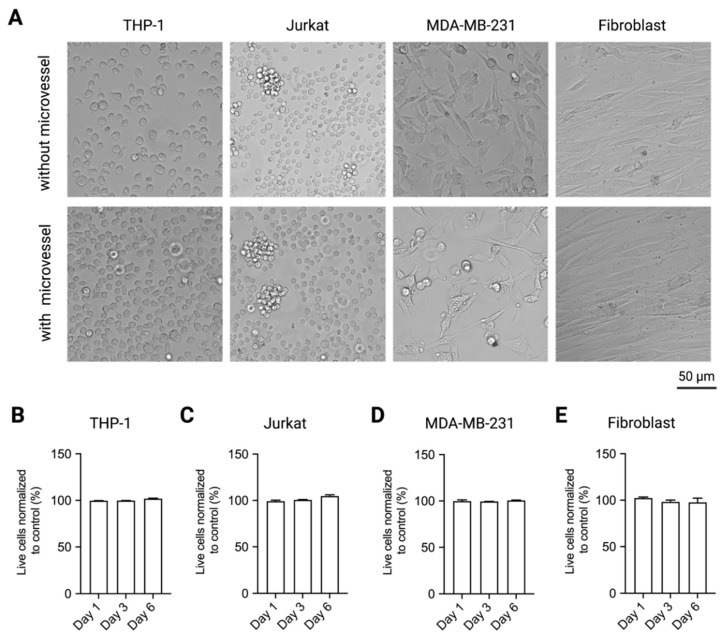
Long-term effects of microvessel on cell culture. (**A**) Images of cells cultured with and without microvessels for 6 days. Long-term viability of (**B**) THP-1, (**C**) Jurkat, (**D**) MDA-MB-231 and (**E**) Fibroblast cells cultured with microvessels for 1 day, 3 days and 6 days. Data are presented as mean ± SD. Experiments were performed in at least 4 replicates.

**Figure 7 ijms-22-06331-f007:**
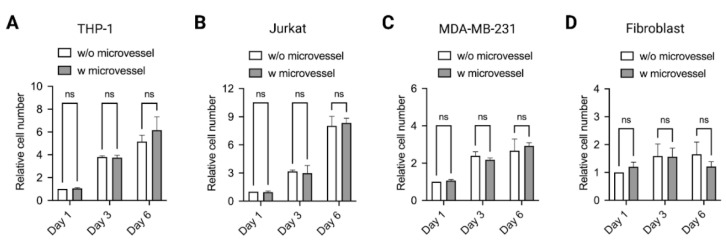
Cell proliferation within microvessel. Proliferation of (**A**) THP-1, (**B**) Jurkat, (**C**) MDA-MB-231 and (**D**) Fibroblast cells cultured for 6 days with and without microvessels. Data are presented as mean ± SD. Experiments were performed in at least 3 replicates. n.s. indicates non significant.

**Table 1 ijms-22-06331-t001:** Primer sequences.

Gene	Forward Sequence	Reverse Sequence
*HIF1A*	TGGATGATGACTTCCAGTTACG	GTGGCAGTGGTAGTGGTGG
*ACTB*	CATCCGCAAAGACCTGTACG	CCTGCTTGCTGATCCACATC

## Data Availability

Details and files to reproduce the fabrication of the microvessels used in this work are available upon request.

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
