# Peer review of "Engineered Microvessel for Cell Culture in Simulated Microgravity"

_ijms, 2021, doi:10.3390/ijms22126331_

Round 1
Reviewer 1 Report
I completely agree that ground-based studies aiming to simulate microgravity conditions on Earth and thereby supporting optimal preparation of space experiments are very important. In addition, hardware developments in order to optimize cultivation conditions and experimental output are necessary.
The authors focus their manuscript on high-throughput, which is their motivation for the development of microvessels for ground-based facilities. 3 D printed vessels made of 4 different materials were tested with respect to biocompatibility with adherent and suspended cells. Based on their results (proliferation, cell viability, stable pH and others) one candidate material was chosen. They suggest the application in ground-based facilities for simulation of microgravity and focus on one of the facilities, the Random Positioning Machine. Random Positioning is one method applied aiming to achieve simulated microgravity conditions. It´s value with respect to biotechnology have been shown, such as the formation of organoids under this condition. Nevertheless, there is a critical discussion concerning the induction and effects of shear forces which are induced due to the randomly changing speed and direction of the rotation of the system and the radius of the exposed samples (Herranz et al.; Hauslage et al. npj.) In contrast, another facility, the 2 D clinostat, has been proven as low-shear stress environment. What I am missing in the manuscript is a critical discussion on sample size and arrangement with respect to the quality of simulation. Hauslage, J. et al., ; Leguy, C. et al., Wuest et al., demonstrated and calculated shearing forces on the RPM. Thus, the limitation is not only high-throughput, which is the motivation for the authors, but the optimization and reduction of non-gravitational effects such as shearing forces during operation in existing set-ups. Thus, the paper needs a critical discussion on the arrangement of their vessels, size and radius with respect to the rotation axis and a deeper look at the mechanical artefacts which occur Why don't you arrange them exactly along the axis of rotation to prevent centrifugal forces which deteriorate the microgravity simulation? Furthermore, why don't you align the vessels in a 2 D clinostat (and explain it´s advantages, see Hauslage et al., Shinde et al., or suggest to use the RPM in a 2D clinostat mode to gain comparative results? Regarding vessels with different diameter distances from the center might be used for threshold studies and can only be classified in the field of tissue engineering and not microgravity simulation.
Furthermore, the scientific value of the manuscript should be increased if the microvessels are validated by means of a cell line and a parameter with has been changed under real microgravity conditions.
Minor suggestions:
Line 39: How do you define a 1D clinostat?
L 48: 2 D clinostat arrangement include small culture systems such as pipettes, thus your assumption “ large vessels” cannot be generalized.
L 68: whether gravitational forces can be neglectable – as you state – depends on the mode of operation (clinorotation versus random positioning).
Fig. 1, microvessels are hard to see.
Line 46: The usage of small vessels on a ground-based facility guarantees a high quality of sim. µg. Larger vessels produce a high amount of residual forces due to a higher acceleration in the outer regions.
Overall comment: This manuscript lacks in the serious discussion about residual forces and the theory behind the simulation of microgravity. When micro-vessels are used as suggested, they should be placed in the center of rotation, doesn’t matter if a RPM or a clinostat is used. Theoretically only in the rotation center (maximum of 4mm diameter) a high quality of simulated microgravity is produced (see literature) which is exceeded in your configuration. The current status of your manuscript provides a suggestion for a new hardware, but no validation with respect to a space experiment. How can this hardware be used by scientists? Is your aim to sell it or more user guide for self-fabrication?
In the end, the manuscript provides a suggestion for performing experiments, which is not enough for a scientific journal. For newcomers in space science it is gives the impression use our microvessels in an RPM and you will do an optimal microgravity simulation experiment – which is not the case due to all the artefacts and not compliance the rules of a stress-free microgravity simulation
Decision: Major revision with respect of a critical discussion of the published results and points of criticism addressed
Author Response
Point-by-point response to reviewers comments
We thank the editor and the reviewers again for their critical evaluation of our reviewed manuscript. We will now address the issues and the reviewed manuscript, as detailed point by point below.
Reviewer 1
I completely agree that ground-based studies aiming to simulate microgravity conditions on Earth and thereby supporting optimal preparation of space experiments are very important. In addition, hardware developments in order to optimize cultivation conditions and experimental output are necessary. The authors focus their manuscript on high-throughput, which is their motivation for the development of microvessels for ground-based facilities. 3 D printed vessels made of 4 different materials were tested with respect to biocompatibility with adherent and suspended cells. Based on their results (proliferation, cell viability, stable pH and others) one candidate material was chosen. They suggest the application in groundbased facilities for simulation of microgravity and focus on one of the facilities, the Random Positioning Machine. Random Positioning is one method applied aiming to achieve simulated microgravity conditions. It´s value with respect to biotechnology have been shown, such as the formation of organoids under this condition. Nevertheless, there is a critical discussion concerning the induction and effects of shear forces which are induced due to the randomly changing speed and direction of the rotation of the system and the radius of the exposed samples (Herranz et al.; Hauslage et al. npj.) In contrast, another facility, the 2 D clinostat, has been proven as low-shear stress environment.
1. What I am missing in the manuscript is a critical discussion on sample size and arrangement with respect to the quality of simulation. Hauslage, J. et al., ; Leguy, C. et al., Wuest et al., demonstrated and calculated shearing forces on the RPM. Thus, the limitation is not only high-throughput, which is the motivation for the authors, but the optimization and reduction of non-gravitational effects such as shearing forces during operation in existing set-ups. Thus, the paper needs a critical discussion on the arrangement of their vessels, size and radius with respect to the rotation axis and a deeper look at the mechanical artefacts which occur Why don't you arrange them exactly along the axis of rotation to prevent centrifugal forces which deteriorate the microgravity simulation? Furthermore, why don't you align the vessels in a 2 D clinostat (and explain it´s advantages, see Hauslage et al., Shinde et al., or suggest to use the RPM in a 2D clinostat mode to gain comparative results? Regarding vessels with different diameter distances from the center might be used for threshold studies and can only be classified in the field of tissue engineering and not microgravity simulation.
Furthermore, the scientific value of the manuscript should be increased if the microvessels are validated by means of a cell line and a parameter with has been changed under real microgravity conditions.
Response
We thank the reviewer for the comments. The scope of our paper is focused on the fabrication and biocompatibility of our cell culture microvessels for use on the simulated microgravity platform. The mode usage of simulated microgravity platforms was not the focus of our work. All experiments (except the liquid weight experiments) were performed on ground (1g) in standard laboratory conditions. However, the only instance when we used the RPM, we made sure that our samples were placed as close to the center of rotation as possible where, according to the user manual of the RPM we are using, the centrifugal forces are minimal. We added this clarification to the manuscript in Line 135 on page 4. Based on literature (Wuest et al.), shear forces appear in large vessels (e.g. T25 flasks). In general, we would expect smaller sized vessels to experience less shear forces due to the lower gradient of acceleration across the vessel. However, numerical simulations should be carried out to further confirm this phenomenon. This was added to the manuscript for clarification in Line 72 on page 2. With respect to the validation using a cell line and a parameter that has been changed under real microgravity, this falls outside the scope of our paper. This is a good point and should be addressed when performing in-depth cell experiments on simulated microgravity platforms which is part of our future work.
Minor suggestions:
2. Line 39: How do you define a 1D clinostat?
Response
We removed this for clarity in Line 40 on page 1. In literature both 1-D and 2-D are used interchangeably but 2-D is more common (Herranz et al. 2013).
3. L 48: 2 D clinostat arrangement include small culture systems such as pipettes, thus your assumption “ large vessels” cannot be generalized.
Response
We thank the reviewer for this comment. Pipettes are limited to suspension cells and generally use volumes larger than 500ul (Herranz et al 2013). We acknowledge that there are small vessels used but “primarily” larger ones are used (Beck et al. 2012, Buken et al. 2019, Martinez et al. 2014, Shi et al. 2020). We provide this discussion in Line 46 on page 2.
4. L 68: whether gravitational forces can be neglectable – as you state – depends on the mode of operation (clinorotation versus random positioning).
Response
We thank the reviewer for this comment. We apologize for this mistake. We changed gravitational forces to “centrifugal forces” for more clarification in Line 70 on page 2.
5. Fig. 1, microvessels are hard to see.
Response
Larger images of the microvessels are shown in Figure 4 and Figure 5. Figure 1 intends to show the overall size of the system. No amendments were performed for this point.
6. Line 46: The usage of small vessels on a ground-based facility guarantees a high quality of sim. μg. Larger vessels produce a high amount of residual forces due to a higher acceleration in the outer regions.
Response
We thank the reviewer for this comment and we agree with it. Line 63 on page 2 addresses this point that small vessels are more optimal for simulated microgravity research, which supports the need for the microvessels proposed in this work.
7. Overall comment: This manuscript lacks in the serious discussion about residual forces and the theory behind the simulation of microgravity. When micro-vessels are used as suggested, they should be placed in the center of rotation, doesn’t matter if a RPM or a clinostat is used. Theoretically only in the rotation center (maximum of 4mm diameter) a high quality of simulated microgravity is produced (see literature) which is exceeded in your configuration. The current status of your manuscript provides a suggestion for a new hardware, but no validation with respect to a space experiment. How can this hardware be used by scientists? Is your aim to sell it or more user guide for self-fabrication?
Response
The aim of this work is not to provide user guidance for simulated microgravity platforms, but to design a small biocompatible cell culture vessel to perform experiments on most simulated microgravity platforms. Again, we do not assess any cell experiments in simulated microgravity conditions in this work. Since the
microvessel is biocompatible in standard conditions, it will not affect cell culture in simulated microgravity conditions. In addition, there are little to no experiments currently conducted within 4mm diameter on RPMs. To be closer to this restriction, our microvessels can be used with one well at the center of rotation which puts it close to the 4mm diameter restriction. Also, the microvessels can be used for other applications such as regenerative medicine, which is independent of these restrictions. We provide details and files upon request as added in Line 284 on page 9, such that any interested scientist
has access to the microvessels if required.
8. In the end, the manuscript provides a suggestion for performing experiments, which is not enough for a scientific journal. For newcomers in space science it is gives the impression use our microvessels in an RPM and you will do an optimal microgravity simulation experiment – which is not the case due to all the artefacts and not compliance the rules of a stress-free microgravity simulation
Response
We do not claim that our proposed microvessels create an optimal microgravity simulation experiment. The work addresses one limitation of the many that exist with simulated microgravity experiments. We aimed to increase throughput and reduce waste while being consistent with control experiments on ground (1g).
Decision: Major revision with respect of a critical discussion of the published results and points of criticism addressed
Reviewer 2 Report
The manuscript from Mei ElGindi and co-workers on “Engineered Microvessel for Cell Culture in Microgravity” is an interesting paper describing the manufacturing, testing and use of multiwell inserts to be use as microvessels for RPM experiments.
The development of such inserts is expected to be quite welcome in the RPM user community since it indeed addresses a long-standing shortcoming. It is worthwhile to be published taking into account some comments.
1: The title should state “simulated” microgravity.
L.52: The authors mention the use of large vessels for RPM studies. In this respect can these microvessels also reduce the fluid shear forces related to RPM rotation. (see Leguy, C. A., et al. (2011). "Fluid motion for microgravity simulations in a random positioning machine." Gravitational and Space Research 25(1); Leguy, C. A. D., et al. (2017). "Fluid dynamics during Random Positioning Machine micro-gravity experiments." Advances in Space Research 59(12): 3045-3057; and Wuest, S. L., et al. (2017). "Fluid dynamics appearing during simulated microgravity using random positioning machines." PLoS One 12(1): e0170826.)
Related to this: this concept makes use of a open / free fluid surface. How does rotation (change of g vector) impact the fluid surface shape ergo the possible fluid movement in the cell? Please comment.
L.81: Fig.1: indicate left image is from ref. 13 and right current concept.
L.85/92: and later. Authors mentioned ‘resin’. What resin was this. Please provide more details.
Fig. 2B-E but also Fig.7 A-D: please use the full scale of the Y-axis to better display the data. Now there is quite some ‘empty’ height.
Part 2.2 L.116 : some parts of this could/should be transferred to M&M like data on volume / 4-well manufacturer etc.
Fig.4: Maybe in 4A add the 4-well plate in the side view and indicate to where the vessel in fluid filled. Also, although already indicated in the supplemental data: include a general dimention in this image.
Fig. 4B: I do not understand the data from 0.0 to 1g in this graph. There is no 0.0 to 1.0g environment regarding RPMs. One better replace this with a more detailed graph starting from 1.0 to 1.5g. (although 1.5g is far to high for ROM related studies).
L.162: I do not see the added values from the tests where HCl and NaOH are added to the medium. Please clarify the rationale or delete this data.
l.173: Fig 8 should be 5D
Just to have a clear understanding: (nearly) all experiments done in para. 2.3 are done in 1g static tests. Not in RPM? Please clarify.
Fig.7: Please clarify ‘relative cell number”? Relative to what: experiment/control? Also, here ore in M&M state what concentrations / cells/surface area has been used in these studies.
L.214: delete “.”
L.231: Delete last ‘and”
- 317: Supplementary Materials: although the design is not complicated, maybe one could also include the STL files from these inserts in light of the ‘open source engineering’ concept.
L.360: Use lower case letters.
Supplementary Material Fig.S1: This is not an “PDMS mold” but a metal mold used for PDMS products.
Regarding the PDMS molding process: maybe some more details can be provided such that future users can easily make use of this. Are there any specific procedures that need to be followed. Are there specific pitfalls that should be avoided in order to have successful microvessels?
Author Response
Point-by-point response to reviewers comments
We thank the editor and the reviewers again for their critical evaluation of our reviewed manuscript. We will now address the issues and the reviewed manuscript, as detailed point by point below.
Reviewer 2
The manuscript from Mei ElGindi and co-workers on “Engineered Microvessel for Cell Culture in Microgravity” is an interesting paper describing the manufacturing, testing and use of multiwell inserts to be use as microvessels for RPM experiments. The development of such inserts is expected to be quite welcome in the RPM user community since it indeed addresses a long-standing shortcoming. It is worthwhile to be published taking into account some comments.
1. The title should state “simulated” microgravity.
Response
We thank the reviewer for this suggestion and agree to state the “simulated” microgravity in the title.
2. L.52: The authors mention the use of large vessels for RPM studies. In this respect can these microvessels also reduce the fluid shear forces related to RPM rotation. (see Leguy, C. A., et al. (2011). "Fluid motion for microgravity simulations in a random positioning machine." Gravitational and Space Research 25(1); Leguy, C. A. D., et al. (2017). "Fluid dynamics during Random Positioning Machine micro-gravity experiments." Advances in Space Research 59(12): 3045-3057; and Wuest, S. L., et al. (2017). "Fluid dynamics appearing during simulated microgravity using random positioning machines." PLoS One 12(1): e0170826.) Related to this: this concept makes use of a open / free fluid surface. How does rotation (change of g vector) impact the fluid surface shape ergo the possible fluid movement in the cell? Please comment.
Response
We thank the reviewer for this important concern. In general, we would expect smaller sized vessels to experience less shear forces due to the lower gradient of acceleration across the vessel. However, numerical simulations should be carried out to further confirm this phenomenon. With respect to the free fluid surface, this study was not part of the scope of the paper. However, it would be an interesting aspect to study later on with experimental techniques like PIV and CFD simulations.
3. L.81: Fig.1: indicate left image is from ref. 13 and right current concept.
Response
We thank the reviewer for this comment. We added clarification to this figure caption.
4. L.85/92: and later. Authors mentioned ‘resin’. What resin was this. Please provide more details.
Response
The resin used for this work was Dental LT Clear Resin from Formlab. This information is found in the materials and methods in Line 249 on page 9.
5. Fig. 2B-E but also Fig.7 A-D: please use the full scale of the Y-axis to better display the data. Now there is quite some ‘empty’ height.
Response
We thank the reviewer for this comment. We usually use the top space to show statistical significance. We believe the current format is clear to the reader. We are unsure if this is what the reviewer is referring to, but we will consider making changes if they have further concerns.
6. Part 2.2 L.116 : some parts of this could/should be transferred to M&M like data on volume / 4-well manufacturer etc.
Response
We believe that providing the volume is critical for the reader to understand and avoid going back and forth between the M&M and results. The manufacturer of the 4-well plates has been moved to the M&M on Line 268 on page 9 as we agree it is not critical for this part.
7. Fig.4: Maybe in 4A add the 4-well plate in the side view and indicate to where the vessel in fluid filled. Also, although already indicated in the supplemental data: include a general dimention in this image.
Response
In Figure 4C we show fluid filled vessels. If the reviewer strongly believes that the fluid filled images are
necessary, we can color the fluid filled area in Figure 4A. We chose not to add the dimensions in the main
paper because the vessel can be scaled for different sized wells. The dimensions for this specific microvessel
are found in the supplementary data.
8. Fig. 4B: I do not understand the data from 0.0 to 1g in this graph. There is no 0.0 to 1.0g
environment regarding RPMs. One better replace this with a more detailed graph starting from 1.0 to 1.5g. (although 1.5g is far to high for ROM related studies).
Response
We thank the reviewer for this critical comment. We initially showed the scale from 0.0 to 1.5g in case these microvessels were to be used in a reduced gravity platform.We agree that in our work we will only be around 1.0g or slightly higher due to centrifugal accelerations. We clarified this point in the figure caption Line 154 on page 5.
9. L.162: I do not see the added values from the tests where HCl and NaOH are added to the medium. Please clarify the rationale or delete this data.
Response
We thank the reviewer for this comment. By adding HCL and NaOH to the cell culture media the maximal buffer capacity in both basic and acidic conditions can be shown. It serves as a control experiment. No amendments were performed for this point.
10. l.173: Fig 8 should be 5D
Response
Thank you for noticing, this has been corrected.
11. Just to have a clear understanding: (nearly) all experiments done in para. 2.3 are done in 1g static tests. Not in RPM? Please clarify.
Response
All experiments are performed under ground (1g) conditions, not on the simulated microgravity platform. This was clarified in the text in Line 192 and 200 on Page 7.
12. Fig.7: Please clarify ‘relative cell number”? Relative to what: experiment/control? Also, here ore in M&M state what concentrations / cells/surface area has been used in these studies.
Response
Relative cell number is the change in cell number relative to day 1 samples. This was clarified in the text Line 202 on page 7. For all experiments, cells were seeded at 2 x 105 cells/well.This information was added in M&M Line 245 on page 8.
13. L.214: delete “.”
Response
Thank you for noticing this. This has been corrected.
14. L.231: Delete last ‘and”
Response
Thank you for noticing this. This has been corrected.
15. 317: Supplementary Materials: although the design is not complicated, maybe one could also include the STL files from these inserts in light of the ‘open source engineering’ concept.
Response
We thank the reviewer for this suggestion. We will be happy to provide the STL files upon request by any potential users of this technology. This clarification was added in the manuscript text in Line 284 on page 9.
16. L.360: Use lower case letters.
Response
Thank you for noticing this. This has been corrected.
17. Supplementary Material Fig.S1: This is not an “PDMS mold” but a metal mold used for PDMS products.
Response
Thank you for noticing this. This has been corrected.
18. Regarding the PDMS molding process: maybe some more details can be provided such that future users can easily make use of this. Are there any specific procedures that need to be followed. Are there specific pitfalls that should be avoided in order to have successful microvessels?
Response
We thank the reviewer for this suggestion. It is a standard two part mold based on our design. We will be happy to provide more details upon request.
Round 2
Reviewer 1 Report
I understand that the focus of your paper is the fabrication and biocompatibility of your cell culture microvessels for use in simulated microgravity and you concentrate on a Random Positioning Machine. The operational mode and principle which is used aiming to achieve simulated microgravity is not your focus, which due to my opinion cannot be ignored. Otherwise, the title and the paper induce in the readership the expectation, that if your vessels are used, simulated microgravity is achieved. Thus, due to my opinion, you cannot ignore the criticism on the RPM and the applied operational mode. I agree that your vessels might be more useful than T25 flasks with respect to shear forces etc. (here you should cite Hauslage et al. npj and Poon, C. DOI: 10.1002/eng2.12242; Factors implicating the validity and interpretation of mechanobiology studies in simulated microgravity environments). All the experiments were done are in 1g in standard laboratory conditions. You state” However, the only instance when we used the RPM, we made sure that our samples were placed as close to the center of rotation as possible where, according to the user manual of the RPM we are using, the centrifugal forces are minimal”. You added an important point in your comment that this is not the closest location: To be closer to this restriction, our microvessels can be used with one well at the center of rotation which puts it close to the 4mm diameter restriction”. This is very important and should be added as a suggestion/advice in your manuscript: – put the sample in the center and test different operational modes, which can already be done on an RPM (Clinostat mode, one direction instead of Random and so on, see Hauslage, J., Cevik, V., & Hemmersbach, R. (2017) to identify the optimal simulation approach. Pyrocystis noctiluca represents an excellent bioassay for shear forces induced in ground-based microgravity simulators (clinostat and random positioning machine). NPJ microgravity, (1), 1-7.
135 The plates were placed as close to the center of rotation as possible to minimize centrifugal forces. Reviewer reply: please correct placed.
Author Response
Point-by-point response to reviewers comments
We thank the editor and the reviewers again for their critical evaluation of our reviewed manuscript. We will now address the issues and the reviewed manuscript, as detailed point by point below. The responses are highlighted in green in the manuscript.
Reviewer 1
I understand that the focus of your paper is the fabrication and biocompatibility of your cell culture microvessels for use in simulated microgravity and you concentrate on a Random Positioning Machine. The operational mode and principle which is used aiming to achieve simulated microgravity is not your focus, which due to my opinion cannot be ignored. Otherwise, the title and the paper induce in the readership the expectation, that if your vessels are used, simulated microgravity is achieved. Thus, due to my opinion, you cannot ignore the criticism on the RPM and the applied operational mode.
1. I agree that your vessels might be more useful than T25 flasks with respect to shear forces etc. (here you should cite Hauslage et al. npj and Poon, C. DOI: 10.1002/eng2.12242; Factors implicating the validity and interpretation of mechanobiology studies in simulated microgravity environments).
Response
These citations have been added and can be found in line 73 on page 2.
2. All the experiments were done are in 1g in standard laboratory conditions. You state” However, the only instance when we used the RPM, we made sure that our samples were placed as close to the center of rotation as possible where, according to the user manual of the RPM we are using, the centrifugal forces are minimal”. You added an important point in your comment that this is not the closest location: To be closer to this restriction, our microvessels can be used with one well at the center of rotation which puts it close to the 4mm diameter restriction”. This is very important and should be added as a suggestion/advice in your manuscript: – put the sample in the center and test different operational modes, which can already be done on an RPM (Clinostat mode, one direction instead of Random and so on, see Hauslage, J., Cevik, V., & Hemmersbach, R. (2017) to identify the optimal simulation approach. Pyrocystis noctiluca represents an excellent bioassay for shear forces induced in ground-based microgravity simulators (clinostat and random positioning machine). NPJ microgravity, (1), 1-7.
Response
We thank the reviewer for the comments. We added this description of placing the microvessels with one well in the center and testing the different operational modes of the RPM. The changes can be found on lines 226-232 on page 8.
3. 135 The plates were placed as close to the center of rotation as possible to minimize centrifugal forces. Reviewer reply: please correct placed.
Response
This has been corrected and can be found on line 135 on page 4.